# Maternal Polyphenols and Offspring Cardiovascular–Kidney–Metabolic Health

**DOI:** 10.3390/nu16183168

**Published:** 2024-09-19

**Authors:** You-Lin Tain, Chien-Ning Hsu

**Affiliations:** 1Division of Pediatric Nephrology, Kaohsiung Chang Gung Memorial Hospital, Kaohsiung 833, Taiwan; tainyl@cgmh.org.tw; 2Institute for Translational Research in Biomedicine, Kaohsiung Chang Gung Memorial Hospital, Kaohsiung 833, Taiwan; 3College of Medicine, Chang Gung University, Taoyuan 333, Taiwan; 4Department of Pharmacy, Kaohsiung Chang Gung Memorial Hospital, Kaohsiung 833, Taiwan; 5School of Pharmacy, Kaohsiung Medical University, Kaohsiung 807, Taiwan

**Keywords:** polyphenols, cardiovascular disease, gut microbiota, resveratrol, developmental origins of health and disease (DOHaD), hypertension, kidney disease, metabolic syndrome

## Abstract

Background: The convergence of cardiovascular, kidney, and metabolic disorders at the pathophysiological level has led to the recognition of cardiovascular–kidney–metabolic (CKM) syndrome, which represents a significant global health challenge. Polyphenols, a group of phytochemicals, have demonstrated potential health-promoting effects. Methods: This review highlights the impact of maternal polyphenol supplementation on the CKM health of offspring. Results: Initially, we summarize the interconnections between polyphenols and each aspect of CKM syndrome. We then discuss in vivo studies that have investigated the use of polyphenols during pregnancy and breastfeeding, focusing on their role in preventing CKM syndrome in offspring. Additionally, we explore the common mechanisms underlying the protective effects of maternal polyphenol supplementation. Conclusions: Overall, this review underscores the potential of early-life polyphenol interventions in safeguarding against CKM syndrome in offspring. It emphasizes the importance of continued research to advance our understanding and facilitate the clinical translation of these interventions.

## 1. Introduction

Cardiovascular disease (CVD) continues to be the leading cause of mortality globally and frequently coexists with metabolic disorders such as diabetes, obesity, and kidney disease [1]. The presence of multiple conditions from these categories significantly increases the risk of mortality. To address this, the American Heart Association has recently defined cardiovascular–kidney–metabolic (CKM) syndrome, highlighting the interconnected nature of these diseases and their shared underlying mechanisms [2]. CKM syndrome is characterized by the simultaneous presence of CVD, CKD, and metabolic disorders such as diabetes mellitus, obesity, and dyslipidemia. CKM syndrome is staged based on the severity and progression of these individual components, ranging from Stage 0 to Stage 4. The concept underscores the importance of considering these conditions as interconnected rather than isolated, which has significant implications for prevention, diagnosis, and treatment [2].

Common underlying mechanisms link CVD, kidney disease, and metabolic disorders, initiating a harmful cycle when activated early in life [3,4]. This cycle perpetuates the progression of CKM syndrome and significantly increases cardiovascular (CV) morbidity and mortality in adulthood. The detrimental interplay among these conditions highlights the urgent need for targeted interventions that address their interconnected nature and emphasizes the importance of early prevention strategies.

An expanding body of research suggests that adverse environmental conditions during gestation and infancy may increase the vulnerability to adult diseases [3,4,5,6]. It covers all aspects of CKM syndrome [3,4,5,6]. This hypothesis states that the developing fetus adapts structurally and functionally to environmental challenges, which can lead to the development of adulthood diseases—a process known as “developmental programming”. This process then gives rise to the concept of developmental origins of health and disease (DOHaD) [3,4,5,6]. Although the pathophysiological mechanisms underlying CKM programming are not yet fully understood, maternal dietary nutrition plays a role in many aspects of CKM syndrome with developmental origins [7,8,9,10]. A notable example is the Dutch famine study, which found that maternal undernutrition during pregnancy is associated with a higher risk of adult offspring developing coronary heart disease, kidney disease, obesity, hyperlipidemia, and hypertension—all features of CKM syndrome [11]. However, there is currently limited human research that provides evidence on how maternal insults can lead to all aspects of CKM syndrome in both children and adults as this is a relatively new and broad field. Conversely, nutritional intervention during gestation and breastfeeding periods may improve, or even reverse, the adverse effects linked to developmental programming through a process known as reprogramming [9].

Polyphenols are crucial phytochemicals produced naturally by plants and are integral to our diet as nutraceuticals. They exhibit a broad spectrum of health benefits, including anti-obesity and antidiabetic effects, antioxidant and anti-inflammatory properties, and prebiotic effects [12,13,14]. Despite these advantages, establishing a precise, evidence-based reference intake for polyphenols remains challenging due to significant variability in evaluation methods, markers, and endpoints used across studies. Additionally, there have been reports of potential negative effects associated with polyphenols [15]. While extensive research has focused on the health benefits of polyphenols, there is a notable lack of studies investigating the effects of maternal polyphenol supplementation specifically for the prevention of offspring CKM syndrome.

## 2. Materials and Methods

This review aims to consolidate recent findings and highlight the impact of maternal polyphenol supplementation on the developmental programming of CKM syndrome. A thorough literature review was conducted by identifying pertinent studies published in English through scientific databases such as MEDLINE, the Cochrane Library, and Embase. Our research encompasses clinical studies, observational studies, clinical trials, and animal research published between January 2000 and April 2024, with a focus on full-text articles written in English. We included studies that specifically address maternal polyphenol supplementation and its impact on CKM syndrome in offspring. We excluded research that examined offspring outcomes unrelated to CKM syndrome or studies restricted to fetal outcomes alone. Additionally, we reviewed reference lists to identify other relevant sources.

The search utilized relevant keywords associated with “polyphenol”, “flavonoid”, “flavans”, “flavanones”, “isoflavones”, “curcumin”, “resveratrol”, “stilbenes”, “lignans”, “tannins”, “anthocyanins”, “obesity”, “kidney disease”, “dyslipidemia”, “diabetes”, “insulin resistance”, “hyperglycemia”, “hypertension”, “fatty liver”, “metabolic syndrome”, “cardiovascular disease”, “atherosclerosis”, “heart failure”, “stroke”, “gestation”, “pregnancy”, “mother”, “maternal”, “lactation”, “breastfeeding”, “neonatal”, “perinatal”, “developmental programming”, “DOHaD”, “offspring”, “progeny”, and “reprogramming”.

## 3. Polyphenols: Chemistry, Bioavailability, and Health Benefits

Polyphenols are a diverse and widely studied group of phenolic compounds, characterized by the presence of two or more phenyl rings and at least one hydroxyl group [11]. The term “polyphenol” originates from the Greek words “poly”, meaning many, and “phenol”, referring to an aromatic ring with an attached hydroxyl group. These compounds are secondary metabolites extensively distributed throughout the plant kingdom and can be roughly grouped into flavonoids and non-flavonoids. To date, over 8000 distinct phenolic structures have been identified, including approximately 5000 flavonoids [12]. Polyphenols are abundant in plant-based foods, notably in fruits, vegetables, nuts, and whole grains, as well as in beverages like tea, chocolate, red wine, and coffee. Table 1 lists polyphenol-abundant plant foods [16,17,18,19,20].

In this review, we have classified polyphenols based on their chemical structures. As illustrated in Figure 1, the primary flavonoids found in foods include flavonols, flavanones, isoflavones, flavones, flavan-3-ols, and anthocyanins. Non-flavonoid phenolic compounds include diarylheptanoids, phenolic acids, xanthones, stilbenes, lignans, and tannins. This review concentrates on the polyphenols most commonly associated with CKM syndrome. It does not include the chemical structures of individual polyphenols. For more detailed information, readers are encouraged to refer to reviews published elsewhere [12].

### 3.1. Flavonoids

Flavonoids account for approximately two-thirds of the total intake of dietary polyphenols [20]. They are classified into several major classes based on the hydroxylation pattern and variations in the chromane ring. These classes include flavones, flavonols, flavanols, flavanones, isoflavones, and anthocyanins. Flavonoids are recognized for their potential health benefits, including antioxidant, antihyperlipidemic, anti-inflammatory, and cardiovascular protective effects [21].

The key flavones found in foods include apigenin, luteolin, and chrysin [22]. The basic chemical structure of flavones contains two benzene rings connected through a heterocyclic pyrone ring [23]. While flavones exhibit numerous potentially beneficial activities [20], their absorption is generally lower compared to other polyphenols.

Flavonols, which include quercetin, kaempferol, myricetin, and isorhamnetin, may influence CV risk factors such as diabetes, hypertension, and hyperlipidemia, potentially reducing CVD mortality. However, data on these effects remain inconsistent [24]. Quercetin, found primarily in apples, onions, and berries [25], has been noted for its anti-hypertensive properties [26].

Flavanols, also known as flavan-3-ols or catechins [27], are mainly found in cocoa, dark chocolate, and berries. Flavanols such as epicatechin and epigallocatechin can be esterified with gallic acid to form compounds like epicatechin gallate (ECG) and epigallocatechin gallate (EGCG) [27]. The presence of flavanols in cocoa and chocolate has garnered interest for their potential role in preventing CVD and hypertension [28]. Flavanones, primarily found in citrus fruits [29], include hesperetin, naringenin, and eriodictyol. High intake of flavanones has been associated with a reduced risk of obesity and diabetes [30].

Isoflavones are predominantly found in legumes, particularly soybeans and soy products [31]. Key isoflavones include genistein, daidzein, and glycitein, among others [32]. These compounds are recognized for their potential health benefits, particularly in relation to hormone-related conditions. The chemical structure of isoflavones enables them to bind to estrogen receptors, potentially leading to both estrogenic and antiestrogenic effects [32]. However, their physiological activity as estrogens in humans remains unclear.

Anthocyanins are glycosides of anthocyanidins, which contribute to their role in giving color to fruits and vegetables [33]. Specifically, cyanidin, delphinidin, malvidin, and pelargonidin are widely distributed in plants [34]. Similar to other flavonoids, anthocyanins also improve CV health [33,34].

### 3.2. Non-Flavonoids

Phenolic acids are a category of organic compounds characterized by aromatic rings linked to a carboxylic acid group. Table 1 highlights various plant-based foods and beverages, including wheat, fruits, red wine, and coffee, which are significant sources of these compounds. The antioxidant properties of phenolic acids are linked to their protective effects against CVD [35,36]. These acids are classified into two main groups: hydroxybenzoic acids (e.g., gallic acid and ellagic acid) and hydroxycinnamic acids (e.g., caffeic acid and coumaric acid).

Xanthones are primarily found in certain plants, especially in mangosteen fruit [37]. This distinctive structure is associated with antioxidant, anti-inflammatory, and CV-protective properties [38].

Stilbenes, a small family of phenylpropanoids, are formed by various plant species [39]. Resveratrol, one of the most extensively studied stilbenes, is commonly found in grapes and red wine [40,41]. Human and animal studies suggest that resveratrol could be an effective polyphenol for the prevention and treatment of obesity, metabolic disorders, diabetes, cardiovascular diseases, and kidney disease [42,43,44,45].

Lignans are phenolic compounds that are commonly found in their glycoside form [46]. Similar to other phenolic compounds, studies have shown that lignans may contribute to CV health by exerting antioxidant and anti-inflammatory effects [47].

Tannins are water-soluble, high molecular weight polyphenolic compounds found in many plants. Tannins are classified into two main groups: hydrolysable and non-hydrolysable. Hydrolysable tannins include gallotannins and ellagitannins, while non-hydrolysable tannins, or proanthocyanidins, are flavonoid polymers common in foods [48]. Tannins offer protection against various biotic and abiotic stressors and have pharmacological effects, including antihypertensive properties [48,49].

Another phenolic compound, diarylheptanoids, is characterized by a structure consisting of two aromatic rings linked by a seven-carbon chain [50]. These compounds are recognized for their broad health-promoting properties and are used as nutraceuticals [50,51]. Curcumin, a notable diarylheptanoid, has been extensively researched for its potential to protect against various diseases, including CVD, metabolic disorders, and kidney disease [52,53,54,55].

Notably, despite initially promising preclinical data suggesting significant health benefits, certain polyphenols, such as curcumin and resveratrol, most studies have failed to demonstrate the same level of effectiveness or impact in human populations. This discrepancy underscores the need for more rigorous and comprehensive studies to better understand the potential and limitations of these polyphenols.

### 3.3. Bioavailability of Polyphenols

Figure 2 illustrates the metabolic fate of dietary polyphenols. Their bioavailability is influenced by several factors, including the physical and chemical properties of the natural matrix, stability during digestion, the action of intestinal enzymes, and interactions with gut microbiota [18,56].

Most polyphenols are released from their conjugates in the food matrix during digestion. During the gastric phase, the action of pepsin, peristaltic movements, and the acidic environment contribute to the partial breakdown and dissolution of polyphenols [57,58]. The low pH may also help polyphenols move from the food matrix to the aqueous phase by reducing interactions between ionic groups.

Only about 5–10% of polyphenols are directly absorbed in the small intestine after undergoing deconjugation processes such as deglycosylation [57]. The bioavailability of polyphenols varies: isoflavones and gallic acid are among the most readily absorbed, while anthocyanins are the least absorbed [57]. Polyphenols can cross the intestinal epithelium via various transport mechanisms, including active transport, passive diffusion, or facilitated diffusion. Low molecular weight polyphenols, such as phenolic acids, are primarily absorbed by passive diffusion [58].

A critical aspect of polyphenol chemistry that impacts digestion and absorption is their strong hydrogen bonding and hydrophobic interactions with biomolecules, especially proteins. For example, tannins bind to proteins, which can hinder their digestion and absorption [59]. This binding tendency is not confined to tannins alone; polyphenols, in general, exhibit a propensity to tightly bind with proteins under favorable conditions. This is due to their phenolic groups, which form robust hydrogen bonds, and their hydrophobic aromatic rings, which facilitate tight protein interactions.

The limited uptake of polyphenols is partly due to their strong interactions with biomolecules, which hinder diffusion and active transport. Once absorbed, polyphenols often bind to serum albumin and other blood proteins, facilitating tissue transport and extending their half-life compared to direct liver metabolism [46,57,58]. In the liver and enterocytes, polyphenols are metabolized into water-soluble conjugates through conjugation reactions with glucuronide, methyl, or sulfate groups. These conjugates enter systemic circulation, reach target organs, and are ultimately excreted in urine. The percentage of polyphenols recovered in urine varies significantly, with isoflavones and gallic acid being the most prominently recovered (over 60%) [46,57]. Polyphenol bioavailability is affected by their intestinal absorption and bioaccessibility, with most polyphenols showing low bioavailability due to poor bioaccessibility [57].

Approximately 90% of ingested polyphenols reach the large intestine, where they have the potential to be transformed into bioavailable products by the resident microbiota [60]. These metabolites are typically recycled through enterohepatic circulation. Gut microbes degrade polyphenols into low molecular weight metabolites that can have similar biological effects as the original compounds [61]. This relationship is reciprocal: microbiota enhance polyphenol bioavailability by converting them into microbial metabolites, while polyphenols can act as prebiotics, influencing gut microbiota composition and supporting beneficial bacteria [61].

## 4. Polyphenols and CKM Syndrome

Epidemiological studies support the associations between polyphenol-abundant diets and reduced risk for chronic diseases [14]. Growing evidence suggests polyphenol-abundant foods and beverages can have antioxidant, anti-inflammatory, anti-obesity, and anti-aging properties and reduce the risk of CKM syndrome-related disorders such as CVD, diabetes mellitus, hypertension, kidney disease, and fatty liver. The following sections explore each of these aspects in detail.

### 4.1. Cardiovascular Disease

Endothelial dysfunction is marked by a state of vasoconstriction and inflammation that contributes to atherosclerosis, the precursor to CVD [62]. This dysfunction impairs endothelium-dependent vasodilation by reducing NO availability [43]. Consuming polyphenol-rich foods such as berries, red wine, chocolate, and green tea can enhance NO production and improve endothelial function [63]. Oxidative stress, inflammation, and lipid deposition are key factors in the progression of endothelial dysfunction and atherosclerosis [62,64]. Polyphenols possess antioxidant and anti-inflammatory properties and can influence lipid metabolism [65]. Specific polyphenols, including quercetin and resveratrol, have been shown to reduce LDL oxidation and cholesterol synthesis, enhance the expression/activity of LDL receptors, and lower the expression of cholesterol transporters [66,67,68].

Additionally, polyphenols may directly affect cardiac rhythm and function by modulating signaling pathways that regulate ion channel activity and cardiac excitability [69,70,71,72]. Therefore, resveratrol, ECG, and EGCG may possess antiarrhythmic properties [69,70,71,72]. Numerous polyphenol-abundant foods and beverages, including tea, cocoa, and grapes, have been investigated for their potential antihypertensive effects [73]. A systematic review of nine randomized controlled trials (RCTs) indicates that grape seed extract significantly lowers systolic BP [48]. Several protective actions of polyphenols in hypertension have been revealed, including enhancement of endothelial function, inhibition of oxidative stress, improvement of NO availability, and inhibition of the renin–angiotensin system (RAS) [74,75]. Despite growing evidence linking polyphenol intake with a reduced risk of CVD, a systematic review has found that only flavonoids—not total polyphenols—are associated with a lower risk of CV events and all-cause mortality [14]. Another systematic review of tea consumption concluded that while tea may be beneficial in lowering risks of CVD and all-cause mortality, the strength of evidence was rated as low [76].

### 4.2. Obesity and Diabetes

Diabetes mellitus arises from impaired glucose metabolism and increasing insulin resistance, leading to chronic hyperglycemia. Obesity is a significant risk factor for type 2 diabetes mellitus [77]. Key contributors to hyperglycemia include the digestion and absorption of dietary carbohydrates, reduced glycogen storage, β-cell dysfunction, and peripheral tissue insulin resistance. In contrast, polyphenols offer potential therapeutic benefits by addressing these issues through several mechanisms [78]. They can reduce the intestinal absorption of carbohydrates, regulate enzymes critical to glucose metabolism, and improve β-cell function and insulin secretion [78,79]. Additionally, polyphenols combat inflammation and oxidative stress, both of which have important roles in the development and progression of diabetes [80].

Phenolic acids, flavonoids, and tannins regulate crucial enzymes such as α-glucosidase and α-amylase, which are involved in carbohydrate digestion [78]. Catechins, epicatechins, and various phenolic acids can inhibit glucose transporters [79]. Anthocyanins, curcumin, and coffee phenols help reduce glucose intolerance and manage postprandial glycemia by enhancing the secretion of glucagon-like peptide-1 and glucose-dependent insulinotropic polypeptide as well as improving insulin response [81]. Catechins and epicatechins reduce hepatic glucose output and hyperglycemia by modulating liver enzyme expression, specifically by upregulating glucose-6-phosphatase and downregulating glucokinase [82]. Ferulic acid lowers blood glucose levels by boosting hepatic glycogen production and glucokinase activity [83]. Furthermore, curcumin is emerging as an effective alternative for managing obesity [84].

A recent systematic review encompassing 44 studies suggests that dietary polyphenols hold promise for the prevention and management of obesity, particularly in specific populations. These populations include individuals under 50 years of age, Asian communities, those with obesity-related health issues, and those consuming polyphenols for periods of 3 months or longer at dosages below 220 mg/day [85]. Additionally, another systematic review of 19 cohort studies found that higher tea consumption (e.g., ≥4 cups/day) is linked to a reduced risk of type 2 diabetes mellitus [86]. Together, these findings underscore the potential of polyphenols in addressing both obesity and diabetes.

### 4.3. NAFLD

Non-alcoholic fatty liver disease (NAFLD) is a chronic liver disease characterized by the accumulation of hepatic steatosis. Hyperlipidemia, obesity, and diabetes are frequently associated with the NAFLD [87]. The accumulation of free fatty acids and lipid metabolites within hepatocytes interferes with insulin-mediated cell signaling, thereby initiating the development of NAFLD [87]. Research suggests that polyphenols can be effective in managing NAFLD. For instance, the combination of curcumin and resveratrol has been revealed to reduce triglyceride levels, total cholesterol, and lipid accumulation while alleviating hepatic steatosis in animal models of NAFLD [88]. Pomegranate polyphenols, including anthocyanins, tannins, and flavonoids, help diminish oxidative stress and inflammation associated with NAFLD [89]. Additionally, polyphenol-rich virgin olive oil has been found to counteract inflammation, insulin resistance, and hepatic oxidative stress induced by a high-fat diet, thereby preventing the progression of NAFLD [90]. Notably, resveratrol has been consistently shown to reduce hepatic steatosis by improving insulin sensitivity and lipid profiles in various animal studies [91]. A systematic review examined dietary supplementation with eight different polyphenols across 2173 participants with NAFLD. While individual compounds such as curcumin, catechin, and silymarin showed potential in improving specific aspects of NAFLD, the overall meta-analysis of all RCTs did not demonstrate significant positive changes [92].

### 4.4. Chronic Kidney Disease

Several lines of evidence support the renoprotective effects of polyphenols. Resveratrol, for example, has shown significant benefits in animal models of kidney disease. These benefits include reducing tubulointerstitial damage, oxidative stress, and inflammation, enhancing antioxidant activity, and improving kidney function [93]. Since acute kidney injury (AKI) is a major cause of CKD, resveratrol has been shown to prevent the transition from AKI to CKD and demonstrates notable renoprotective effects [94]. Flavonoids also play a crucial role in preventing and managing CKD by mitigating oxidative stress and inflammatory pathways [95]. Similarly, EGCG has demonstrated protective effects by reducing inflammation and oxidative stress in a mouse model of unilateral ureteral obstruction [96]. Additionally, compounds such as curcumin, salvianolic acid A, and ellagic acid have shown protective effects in various animal models of CKD [97]. A systematic review supports polyphenol-rich interventions for cardiovascular risk in hemodialysis patients [98], but their impact on CKD progression is still unclear. Another review of 32 studies found that resveratrol lowered creatinine levels and increased GFR, suggesting a mild renal protective effect with low certainty [99].

Although the evidence supports the role of polyphenols in various aspects of CKM syndrome, it also highlights a research gap between preclinical data and human trials. Notably, CKM syndrome is a relatively new and broad concept, and there is limited research investigating the impact of polyphenols on its different characteristics or stages within a single study. Further investigation is needed to address these gaps.

## 5. Polyphenols as Reprogramming Strategies for the Prevention of CKM Syndrome

Although polyphenol supplementation during pregnancy has been evaluated for its potential benefits to both pregnant women and fetal outcomes [100,101,102], there is limited information on its long-term effects on their children, especially regarding CKM syndrome. Given the promising health benefits of polyphenols and their clinical use as nutraceuticals, it is not surprising that these supplements have been evaluated during gestation and lactation in animal models to explore their impact on the developmental programming of CKM syndrome.

Among polyphenols, resveratrol is the most extensively studied in the DOHaD research filed. Table 2 summarizes experimental evidence on the use of non-resveratrol polyphenols for the prevention of offspring CKM syndrome, with a specific focus on early-life interventions during gestation and breastfeeding periods [103,104,105,106,107,108,109,110,111,112]. Studies that specifically address resveratrol are presented separately in Table 3 [113,114,115,116,117,118,119,120,121,122,123,124,125]. Notably, only studies that evaluate offspring outcomes post-weaning have been included in this review.

Rodents are the most commonly used animal models in research. Studies have investigated the effects of polyphenol supplementation in rats across various ages, from 4 to 45 weeks. This range corresponds to different stages of human development, from infancy to adulthood [126]. However, there is limited information on the long-term impact of early polyphenol supplementation on the health of offspring as they age.

### 5.1. Flavonoids

As detailed in Table 2, a range of polyphenol groups—including flavonols [103,104], flavanols [105,106,107], lignans [108], tannins [109,110,111], and diarylheptanoids [112]—have been investigated in animal models of CKM programming. Among the phenotypes associated with CKM syndrome, hypertension is the most frequently studied [104,105,107,111,112], followed by obesity [104,109,110,112] and kidney disease [103,106,108].

Quercetin, a prominent flavonoid, is recognized for its diverse biological activities, including antihypertensive, antihyperlipidemic, antihyperglycemic, antioxidant, anti-inflammatory, and cardioprotective effects [127]. According to Table 2, maternal supplementation with quercetin has been revealed to offer protective benefits to adult offspring, mitigating risks of kidney disease, hyperglycemia, insulin resistance, obesity, and hypertension. This protective effect has been observed in rat models subjected to maternal protein restriction, combined with a high-fructose diet post-weaning, as well as in mouse models with a maternal high-fat diet [103,104].

EGCG, a prominent flavanol, has demonstrated significant potential in preventing metabolic syndrome [128]. In a study using an antenatal dexamethasone exposure model [105], gestational supplementation with EGCG effectively prevented hypertension in the offspring. Green tea, known for its high content of flavanols such as EGCG, epigallocatechin, and epicatechin gallate, has also shown protective effects. Supplementation with green tea extract during lactation safeguarded adult rat offspring from kidney disease when a maternal protein restriction plus post-weaning high-fat diet model was employed [106]. Furthermore, green tea extract administered during gestation and lactation, coupled with a high-fat diet given to offspring after weaning, helped prevent obesity [107].

### 5.2. Non-Flavonoids

Additionally, secoisolariciresinol diglucoside (SDG), the most prevalent lignan, has been observed to protect against kidney disease programmed by a high-fat diet in offspring [108]. Maternal administration of tannins has also been associated with protective effects against offspring obesity, fatty liver, and hypertension [109,110,111]. Grape seed proanthocyanidins extract has been shown to be particularly effective in preventing obesity [129], and its protective effect extends to offspring in a maternal high-fat diet model [109].

Azuki beans, rich in polyphenols such as tannins, quercetin, and phenolic acids, have been found to protect against obesity and fatty liver in offspring complicated by maternal protein restriction during lactation [110]. Moreover, Vitis vinifera L. grape skin extract, which contains tannins, has been shown to protect against hypertension in offspring of dams fed a high-fat diet during lactation [111].

Curcumin, another polyphenol with established CV protective effects [130], has shown some promise in one study for protecting against obesity and hyperlipidemia in adult mouse offspring whose dams were fed a hyperglycemic diet [112]. However, there is a notable gap in some studies regarding the detailed reporting of the types and quantities of active polyphenols administered to animals. For instance, one study utilized grape skin extract (ACH09), which contained approximately 30% polyphenols [111], but did not specify the major active polyphenols involved. This lack of detailed information raises questions about the extent to which different polyphenols contribute to protective effects and highlights the need for further research to identify and quantify the key active components.

Moreover, the reprogramming effects of different polyphenols on offspring health are still largely unexplored. Comprehensive studies are needed to elucidate how different polyphenols impact CKM outcomes in offspring and to clarify their mechanisms of action.

### 5.3. Resveratrol

Among polyphenols, resveratrol is the most common one being studied [113,114,115,116,117,118,119,120,121,122,123,124,125]. Table 3 illustrates that resveratrol supplementation in early life protects adult offspring against almost all aspects of CKM syndrome in various rodent models. There are a number of environmental insults inducing various CKM phenotypes, such as maternal nutritional imbalance [113,117,118,119,120,121,125], maternal illness [114,115,123], and chemical exposure [116,122,123]. Resveratrol was most frequently administered through drinking water at a concentration of 50 mg/L [113,114,115,116,118,119,120,121,122,123], followed by supplementation in chow diets at a concentration of 2–4 g/kg [124,125]. There is limited research on the effects of resveratrol in large animals in the study of CKM syndrome with developmental origins. However, two studies in nonhuman primates have demonstrated that resveratrol supplementation during pregnancy can enhance maternal and placental health and positively impact the fetal liver in mothers consuming Western diets [131,132]. Nonetheless, the long-term outcomes of CKM syndrome in these cases remain undetermined.

Despite the well-documented benefits, the low bioavailability of resveratrol remains a significant obstacle in translating preclinical research into clinical applications [41]. To overcome this limitation, resveratrol derivatives have been developed through chemical modifications aimed at enhancing bioavailability and biological activity.

One approach involves esterifying resveratrol with short-chain fatty acids (SCFAs), resulting in resveratrol SCFA esters designed to improve therapeutic efficacy [133]. Studies have shown that these esters can ameliorate conditions such as hyperlipidemia, obesity, fatty liver, hypertension, and kidney disease in offspring subjected to various early-life adversities [133,134,135,136]. Although the advantageous effects of resveratrol derivatives on specific features of CKM syndrome are promising, further research is needed to evaluate their potential as reprogramming strategies for preventing CKM-related disorders in offspring and to elucidate the underlying protective mechanisms.

## 6. Protective Mechanisms against CKM Programming

Given that various maternal insults can lead to similar adverse outcomes in CKM syndrome in offspring, and considering that polyphenols can mitigate or even reverse these adverse effects, it is likely that common mechanisms underpin the pathogenesis of CKM with developmental origins. Early-life polyphenol intervention may help prevent these mechanisms, which include oxidative stress, abnormal activation of the renin–angiotensin system (RAS), gut microbiota dysbiosis, and epigenetic modulation. Figure 3 illustrates the relationship between polyphenols and these proposed mechanisms underlying CKM programming in response to early-life insults. The following sections will delve into these mechanisms in detail.

### 6.1. Oxidative Stress

Polyphenols help neutralize free radicals—unstable molecules that can induce oxidative stress and damage cells, proteins, and DNA. They achieve this by scavenging free radicals, enhancing antioxidant defenses, inhibiting NADPH oxidase, and increasing levels of glutathione (GSH) [137,138]. Through these mechanisms, polyphenols can reduce oxidative stress and its associated damage, which is implicated in CKM syndrome [139,140,141,142].

As detailed in Table 2 and Table 3, various environmental insults have been connected to oxidative stress and CKM programming. These include exposure to high-fat diets [104], antenatal dexamethasone [105], maternal CKD [114], antenatal exposure to 2,3,7,8-tetrachlorodibenzo-p-dioxin (TCDD) and dexamethasone [122], and maternal exposure to bisphenol A (BPA) combined with a high-fat diet [123]. Quercetin, EGCG, and SDG have been utilized as antioxidants to alleviate oxidative stress and offer protection against various features of CKM syndrome in adult offspring [104,105,108]. Additionally, supplementation with grape skin tannins during pregnancy and lactation has been shown to protect against hypertension complicated by a maternal high-fat diet while also restoring the reduced activities of key antioxidant enzymes, including glutathione peroxidase, catalase, and superoxide dismutase [111].

Moreover, maternal resveratrol supplementation protects against offspring hypertension induced by maternal CKD, which is linked to a reduction in renal expression of 8-hydroxy-2′-deoxyguanosine (8-OHdG), a biomarker of oxidative DNA damage [114]. Additionally, the role of maternal resveratrol therapy in alleviating oxidative stress is demonstrated by its protective effects against hypertension in adult offspring born to dams exposed to TCDD and dexamethasone [122], as well as BPA in conjunction with a high-fat diet [123].

Impairment of the NO pathway is a key oxidative-stress-related mechanism implicated in CKM syndrome with developmental origins [143]. The endogenous inhibitor asymmetric dimethylarginine (ADMA) can inhibit NO generation [144]. Consequently, reducing ADMA concentrations and enhancing NO availability have been proposed as strategies to prevent CKM programming [143]. Table 3 highlights how polyphenols, through the regulation of the NO pathway, exert protective effects against CKM syndrome with developmental origins, as evidenced in various animal models, including maternal CKD [114], maternal co-exposure to ADMA and TMAO [115], maternal L-NAME administration [121], antenatal co-exposure to dexamethasone and TCDD [122], maternal prenatal bisphenol A and high-fat diet exposure [123], and maternal hypertension [124]. These findings highlight the interplay between the polyphenols and oxidative stress implicated in CKM syndrome with developmental origins.

### 6.2. Activation of the RAS

The RAS is a major hormone cascade playing a crucial role in maintaining homeostasis in CV, metabolic, and kidney functions [145]. Abnormal activation of the RAS is identified as a key factor in the development of CKM syndrome. RAS-based therapies have proven effective in reprogramming the mechanisms involved, thereby alleviating the diverse phenotypes associated with CKM syndrome [145]. Activation of the classical renin–angiotensin system (RAS) pathway—specifically, the angiotensin-converting enzyme (ACE)-angiotensin II (Ang II)-angiotensin type 1 receptor (AT1R) axis—promotes the development of hypertension. In contrast, inhibiting this classical RAS pathway or activating the non-classical RAS pathways has been revealed to prevent hypertension [146].

Consistent with previous findings that the antihypertensive effects of various polyphenols are linked to the regulation of the RAS [147,148,149], maternal resveratrol supplementation has been shown to prevent hypertension in offspring across multiple models, as outlined in Table 3. These include exposure to ADMA and TMAO [115], a high-fat diet [120], and prenatal co-exposure to TCDD and dexamethasone [122]. Offspring hypertension programmed by a maternal high-fat diet was associated with an increased level of Ang I and a reduced level of Ang (1–7) in the plasma [120]. Resveratrol supplementation reversed these alterations and also lowered plasma Angiotensin II (Ang II) levels. Similarly, in the ADMA and TMAO models [115], resveratrol prevented offspring hypertension by decreasing the expression of ACE and AT1R while enhancing components of the non-classical RAS pathway.

Overall, resveratrol’s modulation of RAS signaling appears to favor vasodilation. However, the protective effects of other polyphenols on additional aspects of CKM syndrome remain to be explored.

### 6.3. Gut Microbiota

Considering the complex relationship between polyphenols and gut microbiota [150,151], concerted efforts have been directed toward employing early-life gut microbiota-targeted therapies to prevent various aspects of CKM syndrome in offspring [152,153,154].

Perinatal resveratrol supplementation protected adult rat offspring from maternal CKD-primed hypertension, which was linked to an increased proportion of beneficial microbes such as *Bifidobacterium* and *Lactobacillus*, as well as enhanced microbial richness and diversity [114]. Similarly, in a rat model subjected to a high-fructose diet, maternal resveratrol supplementation averted offspring hypertension, accompanied by alterations in gut microbiota composition [118]. Resveratrol treatment was specifically found to increase the proportions of *Lactobacillus* and *Bifidobacterium* [118]. In a model of maternal L-NAME plus a high-fat diet [121], the beneficial effects of resveratrol against hypertension of developmental origins may be related to its ability to diminish the *Firmicutes*-to-*Bacteroidetes* ratio, a microbial marker linked to kidney disease and hypertension [155]. These findings support the concept that early polyphenol use may act as a prebiotic by reshaping the offspring’s gut microbiome and preventing CKD programming. Notably, resveratrol derivatives not only altered gut microbiota composition but also affected microbially derived metabolites. Resveratrol butyrate ester increased fecal SCFA concentrations and improved the gut barrier, thereby preventing maternal BPA exposure-primed metabolic disruption in male offspring [136].

Given that the bioavailability of polyphenols is significantly influenced by gut microbiota, it is crucial to further explore how gut microbiota impacts polyphenol absorption and efficacy, particularly in the context of protecting against CKD syndrome with developmental origins.

### 6.4. Epigenetic Modification

Epigenetic modifications, including histone modification, DNA methylation, and microRNAs (miRNAs), are critical for regulating gene expression, a key mechanism involved in developmental programming [156]. A variety of polyphenols are potential histone deacetylases (HDAC), Sirtuins (SIRTs), and DNA methyltransferase (DNMT) modulators, which control epigenetic modification [157,158]. DNA methylation patterns have been found to correlate with gut microbiota profiles, which in turn are connected to lipid metabolism and obesity [159]. The intake of foods abundant in polyphenols can shape the gut microbiota and alter miRNA expression concurrently [134].

Resveratrol supplementation during gestation and breastfeeding has been revealed to prevent maternal high-fat diet-primed obesity and metabolic disturbances. This protection is linked to the epigenetic regulation of leptin and its receptor via DNA methylation [160]. The epigenetic effects of resveratrol have also been evaluated in a rat model of a high-fructose diet [118]. Specifically, resveratrol’s beneficial effects on offspring hypertension are associated with increased SIRT expression and the activation of various nutrient-sensing pathways in the adult offspring’s kidneys.

As a SIRT1 activator [161], resveratrol plays a key role in mediating the expression of AMPK and its downstream PPAR target genes, all of which are involved in the developmental origins of CKM syndrome [162]. Targeting AMPK signaling through early-life interventions has been proposed as a strategy to prevent the early-life origins of kidney disease and hypertension [163]. In support of this, resveratrol treatment during gestation and breastfeeding protected against offspring hypertension resulting from maternal co-exposure to L-NAME administration and a high-fat diet [121]. This protective effect is likely mediated via the activation of the SIRT1/AMPK pathway. Similarly, maternal consumption of azuki beans, which are rich in polyphenols such as tannins, quercetin, and phenolic acids, during lactation protected against offspring obesity and fatty liver induced by maternal protein restriction, partly due to the upregulation of AMPK phosphorylation [100].

### 6.5. Others

Given the multifaceted biological roles of polyphenols, additional mechanisms may be involved in their protective effects, such as their influence on inflammation [164]. The accumulation of T cells and pro-inflammatory cytokines plays a crucial role in the pathogenesis of kidney disease and hypertension, often triggered by the aryl hydrocarbon receptor (AhR) signaling pathway [165,166]. Increasing evidence supports the idea that activating AhR through environmental chemical exposure is implicated in CKM programming [167]. For instance, two studies indicated that maternal exposure to TCDD or BPA-induced programmed hypertension coincided with the activation of AhR and kidney inflammation [122,123]. In contrast, resveratrol supplementation during gestation and lactation demonstrated anti-inflammatory effects on offspring hypertension by regulating AhR signaling [122,123].

Although many studies have confirmed the anti-inflammatory effects of various polyphenols in preventing and treating many diseases [164], resveratrol is the only polyphenol that has been specifically examined for its anti-inflammatory action in hypertension with developmental origins. Further research is needed to fully understand the potential of polyphenols in early life, especially their anti-inflammatory properties, to advance therapies aimed at preventing CKM syndrome in offspring by targeting inflammation.

While several molecular mechanisms have been outlined, further research is essential to discover other potential mechanisms. A deeper understanding of the interactions between individual polyphenols and each mechanism, along with their differential protective effects, is key to identifying optimal early-life polyphenol interventions for future clinical applications.

## 7. Conclusions and Perspectives

The growing body of animal evidence supporting the protective role of polyphenol supplementation during gestation and lactation in preventing CKM syndrome in offspring is promising but still insufficient. The most significant unresolved issue remains the lack of human translation. Although numerous clinical trials have investigated the health effects of polyphenol-rich foods, polyphenol extracts, and their derivatives [168,169,170], no information is currently available regarding how polyphenol supplementation in pregnant women might influence their children’s long-term health outcomes, particularly concerning CKM syndrome. Notably, certain negative effects of polyphenols have been reported, including inhibition of digestive enzymes, disruption of gut microbiota, interactions with drug metabolism, interference with hormonal balance, prooxidative activity, and mutagenic effects [15]. These impacts can be particularly harmful to specific vulnerable subpopulations.

Some polyphenols can cross the placenta and be transferred to infants through breast milk. While there are potential benefits, research on the optimal levels for pregnant and breastfeeding women is limited. Given that infants’ gut microbiota are still developing and polyphenols may affect nutrient absorption and metabolism, moderation is key. Pregnant and breastfeeding women, along with parents introducing polyphenol-rich foods to infants, should focus on a balanced diet. More studies are needed on the long-term effects, optimal dosages, and interactions during critical developmental stages.

The effects of polyphenols can vary significantly across different developmental stages, influencing their protective impact on CKM syndrome. Factors such as the duration of therapy and the sensitivity of various organs can lead to distinct reprogramming effects. Therefore, further animal studies are crucial for understanding how exposure to different polyphenols at various developmental stages can protect specific aspects of CKM syndrome later in life and for determining the extent of this protection. Additionally, a major challenge in the clinical use of polyphenols is their limited bioavailability in vivo, which restricts their effectiveness and benefits [56,57,58]. Given the complexity and variability in polyphenol pharmacokinetics, more research is needed to elucidate how these factors influence the varying impacts of different polyphenols on CKM syndrome.

While substantial progress has been made in understanding the benefits of resveratrol in CKM programming, other polyphenols have received less attention regarding their reprogramming effects. This review has primarily focused on flavonols, flavanols, lignans, tannins, stilbenes, and diarylheptanoids. However, more comprehensive research is needed to gain a full understanding of the protective mechanisms of different polyphenols and to evaluate their dose-dependency across different models of developmental programming.

In summary, polyphenols play a meaningful role in promoting CKM health in offspring. Gaining a deeper understanding of the mechanisms underlying CKM syndrome with developmental origins, coupled with the growing acknowledgment of the benefits of early polyphenol use, underscores the need for further human research. Such studies are crucial for translating these findings into clinical practice and ultimately reducing the global burden of CKM syndrome.

## Figures and Tables

**Figure 1 nutrients-16-03168-f001:**
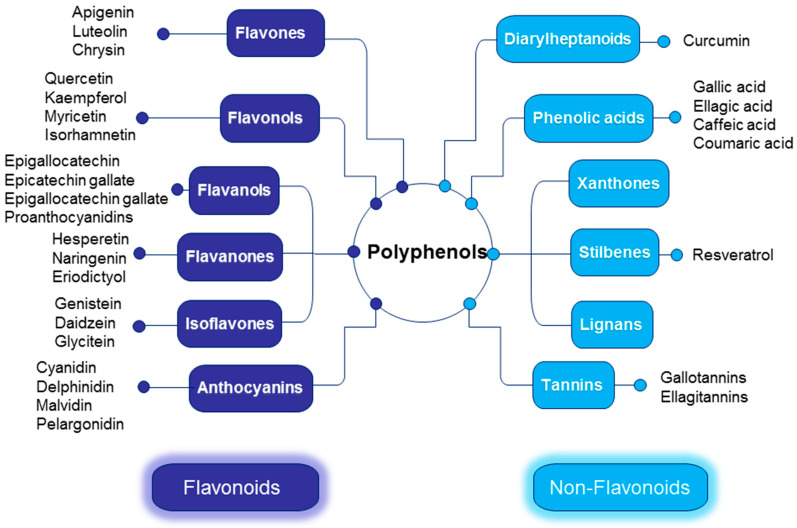
Polyphenol classes and main compounds.

**Figure 2 nutrients-16-03168-f002:**
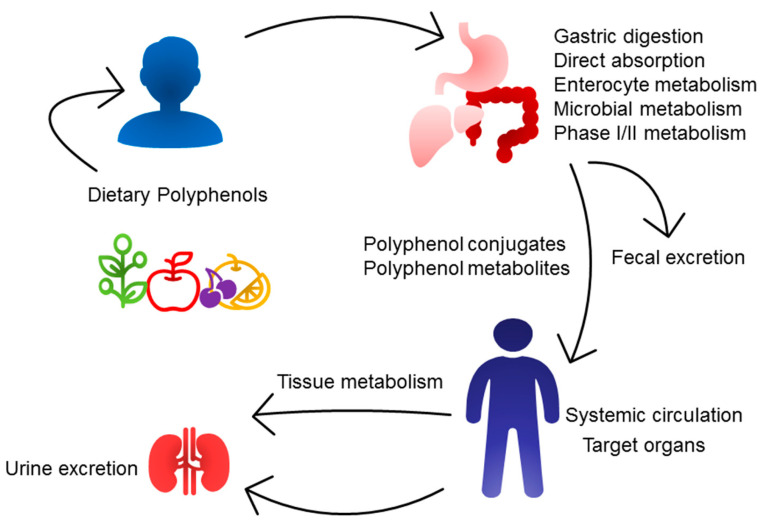
Metabolic fates of dietary polyphenols in the body involve several stages: phase I and II metabolism in the gut and liver, microbial metabolism, absorption into systemic circulation, interaction with target organs, and eventual elimination through feces and urine.

**Figure 3 nutrients-16-03168-f003:**
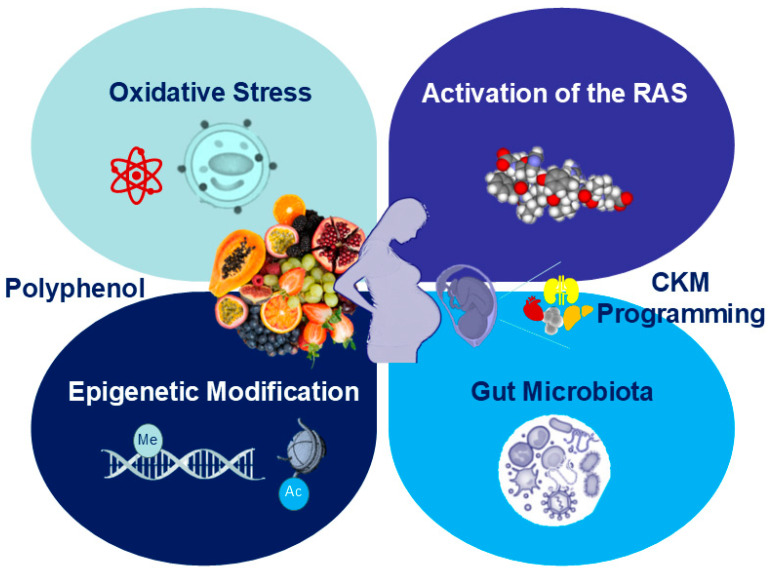
Schema outlining the protective mechanisms of polyphenols behind cardiovascular–kidney–metabolic syndrome with developmental origins.

**Table 1 nutrients-16-03168-t001:** Polyphenol-abundant plant foods.

Plant Food	Comestible Part	Concentrations mg/100 g	Key Polyphenols
Apple	Whole	5–50	Quercetin, phenolic acids
Blueberry	Whole	160–480	Anthocyanins, quercetin, phenolic acids
Coffee	Beverage	90	Phenolic acids
Chestnut	Whole	547–1960	Gallic acid, tannins, ellagic acid
Cacao	Beans	300–1100	Flavonols
Green tea	Extract	29–103	Flavonols
Grapefruit	Flesh	15–115	Flavonoids, phenolic acids
Extra virgin olive oil	Oil	4–200	Lignans, phenolic acids, tannins
Potato	Whole	10–50	Phenolic acids
Plum	Whole	130–240	Anthocyanins, phenolic acids
Red wine	Wine	25–300	Resveratrol, tannins, anthocyanins, phenolic acids
Wheat	Whole	85–220	Phenolic acids
Spinach	Leaf	30–290	Flavonols

**Table 2 nutrients-16-03168-t002:** Animal studies demonstrating polyphenols (excluding resveratrol) prevent offspring CKM syndrome.

Types of Polyphenols	Periods Pregnany/Lactation	Animal Models	Species/ Gender	Age at Evaluation (weeks)	Prevented CKM Phenotypes	Ref.
Flavonols						
Quercetin (0.2% chow)	No/Yes	Maternal PR plus post-weaning high-fructose diet	Wistar rat/F	12	Kidney disease	[103]
Quercetin (50 mg/kg/day)	Yes/No	Maternal high-fat diet	C57BL/6J mouse/M	24	Hyperglycemia, insulin resistance, obesity, and hypertension	[104]
Flavanols						
Epigallocatechin gallate (458 mmol/L) in drinking water	Yes/No	Antenatal dexamethasone exposure	Wistar rat/M & F	14	Hypertension	[105]
Green tea extract (0.24% in chow)	No/Yes	Maternal PR plus post-weaning high-fat diet	Wistar rat/F	45	Kidney disease	[106]
Green tea extract (400 mg/kg/day)	Yes/Yes	Post-weaning high-fat diet	Wistar rat/M	13	Obesity	[107]
Lignans						
Secoisolariciresinol diglucoside (30 mg/kg/day)	Yes/Yes	Maternal and post-weaning high-fat diet	C57BL/6J mouse/M & F	6.5	Kidney disease	[108]
Tannins						
Grape seed procyanidin extract (25 mg/kg/day)	Yes/Yes	Maternal high-fat diet	Rat/M & F	4	Obesity	[109]
Azuki bean (Vigna angularis) polyphenol	No/Yes	Maternal PR	Wistar rat/M	23	Obesity and fatty liver	[110]
Vitis vinifera L. grape skin extract (200 mg/kg/day)	No/Yes	Maternal high-fat diet	SD rat/M	24	Hypertension, hyperlipidemia, hyperglycemia, insulin resistance, and obesity	[111]
Diarylheptanoids						
Curcumin (400 mg/kg/day)	Yes/Yes	Maternal hyperglycemic diet	Swiss mouse/M & F	4	Obesity and hyperlipidemia	[112]

PR = protein restriction; SD rats = Sprague-Dawley rats; M = male; F = female.

**Table 3 nutrients-16-03168-t003:** Animal studies demonstrating resveratrol prevents offspring CKM syndrome.

Dose of Resveratrol	Periods Pregnany/Lactation	Animal Models	Species/ Gender	Age at Evaluation (weeks)	Prevented CKM Phenotypes	Ref.
50 mg/L	Yes/Yes	Maternal high-fat diet	Wistar ras/M & F	3	Obesity, hyperglycemia, and hyperlipidemia	[113]
50 mg/L	Yes/Yes	Maternal CKD	SD rat/M	12	Hypertension	[114]
50 mg/L	Yes/Yes	Maternal ADMA and TMAO exposure	SD rat/M	12	Hypertension	[115]
50 mg/L	Yes/Yes	Maternal TCDD exposure	SD rat/M	12	Hypertension	[116]
25 mg/kg/day	Yes/No	Maternal PR	Wistar ras/M & F	16	Obesity and insulin resistance	[117]
50 mg/L	Yes/Yes	Maternal plus post-weaning high-fructose diet	SD rat/M	12	Hypertension	[118]
50 mg/L	Yes/Yes	Maternal plus post-weaning high-fat diet	SD rat/M	16	Obesity, hyperlipidemia, and hypertension	[119,120]
50 mg/L	Yes/Yes	Maternal L-NAME administration and high-fat diet	SD rat/M	16	Hypertension	[121]
50 mg/L	Yes/Yes	Maternal TCDD and dexamethasone exposure	SD rat/M	16	Hypertension	[122]
50 mg/L	Yes/Yes	Maternal exposure to Bisphenol A and high-fat diet	SD rat/M	16	Hypertension	[123]
4 g/kg of diet	Yes/Yes	Maternal hypertension	SHR/M & F	20	Hypertension	[124]
0.2% in diet	Yes/Yes	Maternal plus post-weaning high-fat diet	C57BL/6 J mouse/M	14	Obesity and hyperlipidemia	[125]

SD rats = Sprague-Dawley rats; SHR = spontaneously hypertensive rat; M = male; F = female.

## Data Availability

Data are contained within the article.

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
