# Peer review of "Maternal Polyphenols and Offspring Cardiovascular–Kidney–Metabolic Health"

_nutrients, 2024, doi:10.3390/nu16183168_

Round 1

Reviewer 1 Report

Comments and Suggestions for Authors

“Maternal polyphenols and offspring….

This review article explores the relationship between maternal ingestion of polyphenols and the health of offspring.

The introduction of the CKM syndrome and DOHaD are brief and provide good overviews.  The introduction to polyphenols needs to be revised.  The text does not reflect the uncertain role of polyphenols in health. Although there are many preliminary studies and correlative studies that suggest polyphenols promote many positive health outcomes, the data remains ambiguous with no carefully designed double blinded studies.  There are several notable cases of exaggerated claims for some compounds that have weakened the entire field—for example resveratrol (omit citation 15). The authors should provide a more balanced view of the role of polyphenols in the early parts of the manuscript, for example using their ref. 61 (a systematic and somewhat skeptical review) to replace one of the overwhelmingly positive reviews 11-15.  Leaders in the field including Paul Kroon and Gary Williamson should serve as the primary basis for the background material to ensure that a balanced viewpoint is presented.

The literature review terms were skewed towards certain types of polyphenols.  For example, the ellagitannins including many specific compounds such as castalagin, vescalagin, etc. are not included—but these are the predominant polyphenols in many berries and nuts.  A more through examination on the polyphenol side would increase the value of the review.  At the very least, the authors should explain why they chose to examine only certain classes of polyphenols.

What were the inclusion criteria (line 83)?  This is key information for understanding the review.  Fig. 1 is not helpful and could be omitted.

In Figure 2, it is incorrect to include the tannin type “proanthocyanidin” under non-flavonoids.  This group of compounds are polymers of the flavan-3-ols such as catechin, epigallocatechin, etc.  The text in lines 169-171 has correct information but the figure is misleading.

In section 3, I suggest omitting all of the chemical structural summaries, which are too brief and are not illustrated with correct formulae—simply refer the reader to an excellent resource such as Stephan Quideau’s definitive review of polyphenol structure and occurrence https://doi.org/10.1002/anie.201000044.

Please provide citations for Table 1, and for claims in the text such as “Flavonoids account for approximately two-thirds of the total intake of dietary polyphenols.” (line 112) or “Quer-124 cetin, found primarily in apples, onions, and berries….” (line 125).  It is particularly important to document primary sources in a review article.  Reference 17 is probably the most useful citation for sources of dietary polyphenols although the USDA database https://www.ars.usda.gov/ARSUserFiles/80400535/Data/Flav/Flav3.3.pdf  is also quite useful.  Idiosyncratic individual compound citations (ref. 16, 18 for example) are less helpful.

Use less dogmatic language when describing the bioactivities of various polyphenols.  For example, although isoflavones do exhibit some estrogenic effects in laboratory settings, it is not clear that they are physiologically active as estrogens in humans.  Change wording for all bioactivities to use language such as “…estrogen receptors, which may allow them to exert both estrogenic…” (line 140) (my change in bold italic font).  There are many sentences that need to be adjusted to indicate uncertainty of bioactivities.  Both resveratrol and cucurmin are examples of compounds whose actual use has not lived up to early claims, and that should be acknowledged in the text.

Figure 3 is not needed, this is a rather simplistic view of the GI tract and has no unique information for polyphenols.

Line 189—What does this mean: “Most polyphenols are generated in the stomach during digestion.”  The sentence suggests that polyphenols are created during digestion.  Perhaps the authors mean “Most polyphenols are released from various conjugates in the food matrix during digestion.”  Please adjust wording.

Line 191-192 suggests that physical degradation is the principle action in the stomach, but polyphenols in general are soluble and would not be “powdered” during the initial phases of digestion.

Line 192-200 needs to be adjusted to emphasize one of the most important facets of polyphenol chemical behavior that impacts digestion and absorption:  The very important hydrogen bonding and hydrophobic interactions between polyphenols and biomolecules in particular protein.  For tannins, interaction with protein impedes their digestion and absorption (e.g. https://doi.org/10.1016/j.bcp.2006.02.013).  The effect is not limited to tannins, since polyphenols of all types bind tightly to protein under favorable conditions.  The combination of excellent hydrogen bonding phenolic groups and highly hydrophobic aromatic rings gives them the distinctive trait of tight binding to biomolecules. 

The low uptake noted for polyphenols is at least in part due to their tendency to interact strongly with other biomolecules, limiting diffusion as well as active transport.  After absorption, polyphenols may adsorb to serum albumin and other blood proteins, resulting in transport to tissues and prolonged lifetime compared to direct metabolism in the liver.  The unique binding character of polyphenols should be highlighted by the authors with relevant references.

What is the source for the claim: “Approximately 90-95% of polyphenols are metabolized by gut microbiota, which are 209 essential for breaking down dietary polyphenols and their phase I/II metabolites” (line 209-210).  I am not familiar with studies supporting that idea, in fact studies of polyphenols and the gut microbiome are still quite novel and there is much to learn.

For Section 4, the authors should find and report on the many recent systematic reviews rather than the bewildering number of individual studies.  The authors should be sure to describe the findings of the reviews carefully to provide a balanced view of polyphenol activities.  For example, a recent systematic review of tea consumption concluded that while tea may be beneficial, the quality of the available data is limited e.g. “strength of evidence was rated as low”.  https://doi.org/10.1093%2Fadvances%2Fnmaa010  The approach of using systematic reviews to identify the research gaps in polyphenol bioactivity studies would make this paper novel.  We do not need more catalogs of supposed beneficial activities—instead we need resources that accurately reflect the state of the field.

It would be helpful to include some of the recent literature that emphasizes negative effects of polyphenols e.g. https://doi.org/10.3390%2Fmolecules28062536 was included in this review.

In section 5, there is a good effort to describe the new area of polyphenols and DOHaD, with specific details and limitations of studies summarized.

Section 6 attempts to summarize possible mechanisms for polyphenol effects.  This is problematic because there are not any well-defined specific effects for polyphenols—instead they seem to be active in every realm of cell control and cell damage.  The mechanisms are “thrown on the wall to see what sticks”—and nothing seems to stick!  I recommend shortening this to a list of possible mechanisms citing a few pertinant review articles.  Add a section about polyphenols as “PAINs”, or Pan-Assay Interference Compounds (also sometimes called nuisance compounds) e.g. searching for antiviral activity among polyphenols https://doi.org/10.3389%2Ffphar.2022.909945

Focusing and balancing the information in this review would make it a useful contribution to the literature. Reducing the amount of information in section 3 and section 4, and cutting section 6 substantially, will help make the article a manageable length.  The figures and table should be carefully evaluated to make sure they provide essential information to support the revised text.

Author Response

RESPONSES TO REVIEWER’S COMMENTS

Reviewer #1

This review article explores the relationship between maternal ingestion of polyphenols and the health of offspring.

RESPONSE: We express our appreciation to Reviewer #1 for his/her meticulous evaluation and valuable insights into our manuscript.

The introduction of the CKM syndrome and DOHaD are brief and provide good overviews.  The introduction to polyphenols needs to be revised.  The text does not reflect the uncertain role of polyphenols in health. Although there are many preliminary studies and correlative studies that suggest polyphenols promote many positive health outcomes, the data remains ambiguous with no carefully designed double blinded studies.  There are several notable cases of exaggerated claims for some compounds that have weakened the entire field—for example resveratrol (omit citation 15). The authors should provide a more balanced view of the role of polyphenols in the early parts of the manuscript, for example using their ref. 61 (a systematic and somewhat skeptical review) to replace one of the overwhelmingly positive reviews 11-15.  Leaders in the field including Paul Kroon and Gary Williamson should serve as the primary basis for the background material to ensure that a balanced viewpoint is presented.

RESPONSE: As suggested, we have revised the following statements and references to present a more balanced perspective on the role of polyphenols.

Lines 63-72: “Polyphenols are crucial phytochemicals produced naturally by plants and are integral to our diet as nutraceuticals. They exhibit a broad spectrum of health benefits, including anti-obesity and antidiabetic effects, antioxidant and anti-inflammatory proper-ties, and prebiotic effects [12-14]. Despite these advantages, establishing a precise, evidence-based reference intake for polyphenols remains challenging due to significant variability in evaluation methods, markers, and endpoints used across studies. Additionally, there have been reports of potential negative effects associated with polyphenols [15]. While extensive research has focused on the health benefits of polyphenols, there is a notable lack of studies investigating the effects of maternal polyphenol supplementation specifically for the prevention of offspring CKM syndrome.”

The literature review terms were skewed towards certain types of polyphenols.  For example, the ellagitannins including many specific compounds such as castalagin, vescalagin, etc. are not included—but these are the predominant polyphenols in many berries and nuts.  A more through examination on the polyphenol side would increase the value of the review.  At the very least, the authors should explain why they chose to examine only certain classes of polyphenols.

RESPONSE: We acknowledge the Reviewer’s point that our review may not cover every class of polyphenols in detail. As indicated in Tables 2 and 3, our examination is limited to specific polyphenol classes relevant to CKM programming. Consequently, the review focuses primarily on the most commonly studied polyphenols associated with CKM syndrome. We appreciate your understanding and hope this addresses your concerns. We have added the following statements for clarity.

Lines 109-112: “This review concentrates on the polyphenols most commonly associated with CKM syndrome. It does not include the chemical structures of individual polyphenols. For more detailed information, readers are encouraged to refer to reviews published elsewhere [12].”

What were the inclusion criteria (line 83)?  This is key information for understanding the review.  Fig. 1 is not helpful and could be omitted.

RESPONSE: As suggested, we have added the following statements to clarify the inclusion and exclusion criteria and have removed Figure 1.

Lines 75-84: “A thorough literature review was conducted by identifying pertinent studies published in English through scientific databases such as MEDLINE, the Cochrane Library, and Em-base. Our research encompasses clinical studies, observational studies, clinical trials, and animal research published between January 2000 and April 2024, with a focus on full-text articles written in English. We included studies that specifically address maternal poly-phenol supplementation and its impact on CKM syndrome in offspring. We excluded re-search that examined offspring outcomes unrelated to CKM syndrome or studies restrict-ed to fetal outcomes alone. Additionally, we reviewed reference lists to identify other relevant sources.”

In Figure 2, it is incorrect to include the tannin type “proanthocyanidin” under non-flavonoids.  This group of compounds are polymers of the flavan-3-ols such as catechin, epigallocatechin, etc.  The text in lines 169-171 has correct information but the figure is misleading.

RESPONSE: We have redrawn Figure 2 to prevent any potential misinterpretation.

In section 3, I suggest omitting all of the chemical structural summaries, which are too brief and are not illustrated with correct formulae—simply refer the reader to an excellent resource such as Stephan Quideau’s definitive review of polyphenol structure and occurrence https://doi.org/10.1002/anie.201000044.

RESPONSE: Per the suggestion, we have omitted all of the chemical structural summaries and referred the readers to the references.

Please provide citations for Table 1, and for claims in the text such as “Flavonoids account for approximately two-thirds of the total intake of dietary polyphenols.” (line 112) or “Quer-124 cetin, found primarily in apples, onions, and berries….” (line 125).  It is particularly important to document primary sources in a review article.  Reference 17 is probably the most useful citation for sources of dietary polyphenols although the USDA database https://www.ars.usda.gov/ARSUserFiles/80400535/Data/Flav/Flav3.3.pdf  is also quite useful.  Idiosyncratic individual compound citations (ref. 16, 18 for example) are less helpful.

RESPONSE: As suggested, we have replaced References #16 and #18 and added new references throughout the section as indicated.

Use less dogmatic language when describing the bioactivities of various polyphenols.  For example, although isoflavones do exhibit some estrogenic effects in laboratory settings, it is not clear that they are physiologically active as estrogens in humans.  Change wording for all bioactivities to use language such as “…estrogen receptors, which may allow them to exert both estrogenic…” (line 140) (my change in bold italic font).  There are many sentences that need to be adjusted to indicate uncertainty of bioactivities.  Both resveratrol and cucurmin are examples of compounds whose actual use has not lived up to early claims, and that should be acknowledged in the text.

RESPONSE: As suggested, we have rephrased the sentence to address this point and have also revised other parts of the text to better reflect the uncertainty regarding bioactivities.

Lines 142-145: “The chemical structure of isoflavones enables them to bind to estrogen receptors, potentially leading to both estrogenic and antiestrogenic effects [32]. However, their physiological activity as estrogens in humans remains unclear.”

We also added the following statements to the discrepancy between preclinical data and human studies.

Lines 181-185: “Notably, despite initially promising preclinical data suggesting significant health benefits, certain polyphenols, such as curcumin and resveratrol, have failed to demonstrate the same level of effectiveness or impact in human populations. This discrepancy underscores the need for more rigorous and comprehensive studies to better understand the potential and limitations of these polyphenols.”

Figure 3 is not needed, this is a rather simplistic view of the GI tract and has no unique information for polyphenols.

RESPONSE: While Figure 3 may seem simplistic to specialists, it is designed to be highly accessible and informative for general readers who may not be familiar with the metabolic pathways of dietary polyphenols. We hope the Reviewer appreciates this perspective and agrees to retain the figure in its current form, as supported by the other reviewers.

Line 189—What does this mean: “Most polyphenols are generated in the stomach during digestion.”  The sentence suggests that polyphenols are created during digestion.  Perhaps the authors mean “Most polyphenols are released from various conjugates in the food matrix during digestion.”  Please adjust wording.

RESPONSE: We have rephrased the sentence as follows.

“Most polyphenols are released from their conjugates in the food matrix during digestion.”

Line 191-192 suggests that physical degradation is the principle action in the stomach, but polyphenols in general are soluble and would not be “powdered” during the initial phases of digestion.

RESPONSE: We have rephrased the sentence as follows:

“During the gastric phase, the action of pepsin, peristaltic movements, and the acidic environment contribute to the partial breakdown and dissolution of polyphenols [57,58].”

Line 192-200 needs to be adjusted to emphasize one of the most important facets of polyphenol chemical behavior that impacts digestion and absorption:  The very important hydrogen bonding and hydrophobic interactions between polyphenols and biomolecules in particular protein.  For tannins, interaction with protein impedes their digestion and absorption (e.g. https://doi.org/10.1016/j.bcp.2006.02.013).  The effect is not limited to tannins, since polyphenols of all types bind tightly to protein under favorable conditions.  The combination of excellent hydrogen bonding phenolic groups and highly hydrophobic aromatic rings gives them the distinctive trait of tight binding to biomolecules.

RESPONSE: As suggested, we have incorporated the following paragraph to address this point.

Lines 207-231: “A critical aspect of polyphenol chemistry that impacts digestion and absorption is their strong hydrogen bonding and hydrophobic interactions with biomolecules, especially proteins. For example, tannins bind to proteins, which can hinder their digestion and absorption [59]. This binding tendency is not confined to tannins alone; polyphenols in general exhibit a propensity to tightly bind with proteins under favorable conditions. This is due to their phenolic groups, which form robust hydrogen bonds, and their hydrophobic aromatic rings, which facilitate tight protein interactions.”

The low uptake noted for polyphenols is at least in part due to their tendency to interact strongly with other biomolecules, limiting diffusion as well as active transport.  After absorption, polyphenols may adsorb to serum albumin and other blood proteins, resulting in transport to tissues and prolonged lifetime compared to direct metabolism in the liver.  The unique binding character of polyphenols should be highlighted by the authors with relevant references.

RESPONSE: We have rephrased the following statements for clarity.

Lines 214-219: “The limited uptake of polyphenols is partly due to their strong interactions with bio-molecules, which hinder diffusion and active transport. Once absorbed, polyphenols often bind to serum albumin and other blood proteins, facilitating tissue transport and extending their lifespan compared to direct liver metabolism [46,57,58]. In the liver and enterocytes, polyphenols are metabolized into water-soluble conjugates through conjugation re-actions with glucuronide, methyl, or sulfate groups.”

What is the source for the claim: “Approximately 90-95% of polyphenols are metabolized by gut microbiota, which are 209 essential for breaking down dietary polyphenols and their phase I/II metabolites” (line 209-210).  I am not familiar with studies supporting that idea, in fact studies of polyphenols and the gut microbiome are still quite novel and there is much to learn.

RESPONSE: We have rephrased the following sentence and provided the relevant reference.

Lines 225-226: “Approximately 90% of ingested polyphenols reach the large intestine, where they are transformed into bioavailable products by the resident microbiota [60].”

For Section 4, the authors should find and report on the many recent systematic reviews rather than the bewildering number of individual studies.  The authors should be sure to describe the findings of the reviews carefully to provide a balanced view of polyphenol activities.  For example, a recent systematic review of tea consumption concluded that while tea may be beneficial, the quality of the available data is limited e.g. “strength of evidence was rated as low”.  https://doi.org/10.1093%2Fadvances%2Fnmaa010  The approach of using systematic reviews to identify the research gaps in polyphenol bioactivity studies would make this paper novel.  We do not need more catalogs of supposed beneficial activities—instead we need resources that accurately reflect the state of the field.

RESPONSE: As suggested, we have added several references to this section to address research gaps and reflect the current state of the field regarding systematic reviews.

It would be helpful to include some of the recent literature that emphasizes negative effects of polyphenols e.g. https://doi.org/10.3390%2Fmolecules28062536 was included in this review.

RESPONSE: Per the suggestion, we have added the following statements into the Discussion to address this aspect.

Lines 591-595: “Notably, certain negative effects of polyphenols have been reported, including inhibition of digestive enzymes, disruption of gut microbiota, interactions with drug metabolism, interference with hormonal balance, prooxidative activity, and mutagenic effects [15]. These impacts can be particularly harmful to specific vulnerable subpopulations.”

In section 5, there is a good effort to describe the new area of polyphenols and DOHaD, with specific details and limitations of studies summarized.

RESPONSE: Again, we thank the Reviewer for their efforts

Section 6 attempts to summarize possible mechanisms for polyphenol effects.  This is problematic because there are not any well-defined specific effects for polyphenols—instead they seem to be active in every realm of cell control and cell damage.  The mechanisms are “thrown on the wall to see what sticks”—and nothing seems to stick!  I recommend shortening this to a list of possible mechanisms citing a few pertinant review articles.  Add a section about polyphenols as “PAINs”, or Pan-Assay Interference Compounds (also sometimes called nuisance compounds) e.g. searching for antiviral activity among polyphenols https://doi.org/10.3389%2Ffphar.2022.909945

RESPONSE: We respectfully ask the Reviewer to consider that Section 6, "Protective Mechanisms against CKM Programming," focuses on the mechanisms by which polyphenols may influence reprogramming rather than their direct effects. The beneficial effects of polyphenols that are already known may differ from their reprogramming effects. Therefore, the data presented in Section 6 provides novel and important insights. We hope the Reviewer understands the significance of this focus and agrees to retain this section in its current form.

While the concept of polyphenols as Pan-Assay Interference Compounds is intriguing, it is currently supported by only a limited number of reports, none of which pertain specifically to the DOHaD research field. We recognize the potential significance of this concept and agree that it warrants further investigation in future studies.

Focusing and balancing the information in this review would make it a useful contribution to the literature. Reducing the amount of information in section 3 and section 4, and cutting section 6 substantially, will help make the article a manageable length.  The figures and table should be carefully evaluated to make sure they provide essential information to support the revised text.

RESPONSE: We sincerely thank the Reviewer for their valuable feedback and efforts to improve our manuscript. We have revised the manuscript to provide more focused and balanced information in response to their suggestions.

Reviewer 2 Report

Comments and Suggestions for Authors

The aim of this manuscript is to review the effects of polyphenols on cardiovascular-kidney-metabolic (CKM) syndrome. The authors explore the relationship between various polyphenols and CKM, followed by an examination of their effects in animal models during pregnancy and breastfeeding. Potential mechanisms underlying these effects are also discussed.

This is an interesting review, and I have several suggestions:

  1. In the introduction, the cardiovascular-kidney-metabolic syndrome should be defined in more detail.
  2. Additionally, the manuscript should clarify how CKM syndrome differs from metabolic syndrome, cardiorenal syndrome, NAFLD/NASH, obesity. More detailed information is needed, especially in the results section, where the effects of polyphenols on cardiovascular disease, CKD, NASH, and insulin resistance are reviewed, but then the authors conclude with the effects of polyphenols on CKM.
  3. A more comprehensive review of polyphenols' effects on cardiovascular disease, NAFLD/NASH, CKD, and insulin resistance should be provided. This could be enhanced with a figure demonstrating the potential effects and mechanisms discussed in section 4 of the results.
  4. The safety of polyphenols for offspring and infants should also be discussed.

Author Response

Reviewer #2

The aim of this manuscript is to review the effects of polyphenols on cardiovascular-kidney-metabolic (CKM) syndrome. The authors explore the relationship between various polyphenols and CKM, followed by an examination of their effects in animal models during pregnancy and breastfeeding. Potential mechanisms underlying these effects are also discussed.

RESPONSE: We thank Reviewer #2 for his/her generous support.

This is an interesting review, and I have several suggestions:

In the introduction, the cardiovascular-kidney-metabolic syndrome should be defined in more detail.

RESPONSE: As suggested, we have added the following statements to describe the CKM syndrome in more detail:

Lines 33-39: “CKM Syndrome is characterized by the simultaneous presence of CVD, CKD, and metabolic disorders, such as diabetes mellitus, obesity, and dyslipidemia. CKM Syndrome is staged based on the severity and progression of these individual components, ranging from Stage 0 to Stage 4. The concept underscores the importance of considering these conditions as interconnected rather than isolated, which has significant implications for prevention, diagnosis, and treatment [2].”

Additionally, the manuscript should clarify how CKM syndrome differs from metabolic syndrome, cardiorenal syndrome, NAFLD/NASH, obesity. More detailed information is needed, especially in the results section, where the effects of polyphenols on cardiovascular disease, CKD, NASH, and insulin resistance are reviewed, but then the authors conclude with the effects of polyphenols on CKM.

RESPONSE: As suggested, we have added the following statements to address this issue.

Lines 331-335: “Although the evidence supports the role of polyphenols in various aspects of CKM syndrome, it also highlights a research gap between preclinical data and human trials. Notably, CKM syndrome is a relatively new and broad concept, and there is limited research investigating the impact of polyphenols on its different characteristics or stages within a single study. Further investigation is needed to address these gaps.”

A more comprehensive review of polyphenols' effects on cardiovascular disease, NAFLD/NASH, CKD, and insulin resistance should be provided. This could be enhanced with a figure demonstrating the potential effects and mechanisms discussed in section 4 of the results.

RESPONSE: Our review primarily focuses on the reprogramming effects of polyphenols rather than their direct impacts. Consequently, Section 6 emphasizes how early-life polyphenol interventions might help prevent CKM syndrome programming. In Section 4, we provide a brief summary of the current evidence on polyphenols' effects on various phenotypes of CKM syndrome for general readers. Based on suggestions, we have added several references regarding systematic reviews to enhance the evidence presented in Section 4. Since Section 6 already includes a figure illustrating these concepts, we believe an additional figure in Section 4 to demonstrate direct effects may not be necessary. We hope the Reviewer agrees with this approach.

The safety of polyphenols for offspring and infants should also be discussed.

RESPONSE: As suggested, we have added the following statements to discuss the safety issue.

Lines 596-602: “Some polyphenols can cross the placenta and be transferred to infants through breast milk. While there are potential benefits, research on the optimal levels for pregnant and breastfeeding women is limited. Given that infants' gut microbiota is still developing and polyphenols may affect nutrient absorption and metabolism, moderation is key. Pregnant and breastfeeding women, along with parents introducing polyphenol-rich foods to infants, should focus on a balanced diet. More studies are needed on the long-term effects, optimal dosages, and interactions during critical developmental stages.”

Reviewer 3 Report

Comments and Suggestions for Authors

In this systematic review?, the authors discussed the impact of maternal polyphenol consumption to the offsprings' CKM health. This can be an insightful manuscript. I do have some comments and suggestions:

1) Please be clear whether this is a systematic review or a narrative review. I saw that the authors' added the search strategy and flowchart but it was not specified that this is a systematic review. Please specify in the title. 

2) If this is a sysrev, please adhere to PRISMA guideline and attach the PRISMA checklist. Otherwise, please remove the "systematic search" and flowchart. 

3) If this is a systematic review, I don't see the table listing all of the included studies. Thus, I am not sure if the writing of this manuscript is appropriate.

4) Section 4.1 does not discuss the potential benefits of polyphenols for cardiac arrhythmias (PMID: 30238135; 26205342) through the modulations of cardiac ion channels (PMID: 28117139; 26205342). This aspect is relevant and must be discussed. 

5) Section 4.4. can include this recent discussion about the benefits of resveratrol on AKI and CKD (PMID: 38662688)

6) As I understand, the offspring benefits are mostly due to the antioxidant activity of polyphenols. If this is true, what is the superiority of these substance over other antioxidants? 

7) What is also missing is the timeline of administration. When should this be given to the mother to prevent CKM disease in the children? Who are the candidates, only high risk mothers with comorbidities or everyone? Please elaborate. 

8) Please provide a table listing the human studies on the topic. This will strengthen the proposed argument that polyphenols are indeed useful. Otherwise, it remains speculative whether this would actually work in human. 

9) CKM is an extremely broad terminology. I think whenever possible, the authors need to specify the details of cardiovascular and kidney diseases explored in each studies and how they induced the disease. For instance, hypertension induced CKD would have different pathophysiology than diabetes induced CKD. Thus, the cellular target of polyphenols may vary. 

10) The review would benefit from a schematic drawing explaining how maternal health can induce CKM disease in the offsprings and which part of the cascade polyphenols would exert their potentials. 

Comments on the Quality of English Language

minor editing is required

Author Response

Reviewer #3

In this systematic review?, the authors discussed the impact of maternal polyphenol consumption to the offsprings' CKM health. This can be an insightful manuscript. I do have some comments and suggestions:

RESPONSE: We sincerely appreciate Reviewer #3 for their thoughtful efforts and constructive feedback on our work.

1) Please be clear whether this is a systematic review or a narrative review. I saw that the authors' added the search strategy and flowchart but it was not specified that this is a systematic review. Please specify in the title.

RESPONSE: This is a scoping review but not a systemic review.

2) If this is a sysrev, please adhere to PRISMA guideline and attach the PRISMA checklist. Otherwise, please remove the "systematic search" and flowchart.

RESPONSE: Following the suggestion, we have removed the search strategy and flowchart.

3) If this is a systematic review, I don't see the table listing all of the included studies. Thus, I am not sure if the writing of this manuscript is appropriate.

RESPONSE: Again, this is a scoping review and does not follow the PRISMA guidelines for systematic reviews.

4) Section 4.1 does not discuss the potential benefits of polyphenols for cardiac arrhythmias (PMID: 30238135; 26205342) through the modulations of cardiac ion channels (PMID: 28117139; 26205342). This aspect is relevant and must be discussed.

RESPONSE: Per your request, we have included the following statements to address this point.

Lines 252-255: “Additionally, polyphenols may directly affect cardiac rhythm and function by modulating signaling pathways that regulate ion channel activity and cardiac excitability [69-72]. Therefore, resveratrol, ECG, and EGCG may possess antiarrhythmic properties [69-72].”

5) Section 4.4. can include this recent discussion about the benefits of resveratrol on AKI and CKD (PMID: 38662688)

RESPONSE: As suggested, we have incorporated the following statements to underscore this aspect.

Lines 319-321: “Since acute kidney injury (AKI) is a major cause of CKD, resveratrol has been shown to prevent the transition from AKI to CKD and demonstrates notable renoprotective effects [94].”

6) As I understand, the offspring benefits are mostly due to the antioxidant activity of polyphenols. If this is true, what is the superiority of these substance over other antioxidants?

RESPONSE: As discussed in Section 6, 'Protective Mechanisms against CKM Programming,' early-life polyphenol intervention offers several protective mechanisms. Beyond their antioxidant activity, polyphenols can protect against offspring CKM syndrome by regulating the renin-angiotensin system (RAS), gut microbiota, and epigenetic factors. These effects may not be solely attributed to their antioxidant properties.

7) What is also missing is the timeline of administration. When should this be given to the mother to prevent CKM disease in the children? Who are the candidates, only high risk mothers with comorbidities or everyone? Please elaborate.

RESPONSE: In this review, we primarily summarize experimental evidence on the use of polyphenols to prevent CKM syndrome in offspring, focusing on early-life interventions during gestation and breastfeeding. However, the safety of polyphenols for normal pregnancy, as well as their effects on offspring and infants, is an important consideration. To address this, we have included a discussion on these safety concerns in the revised text.

Lines 597-603: “Some polyphenols can cross the placenta and be transferred to infants through breast milk. While there are potential benefits, research on the optimal levels for pregnant and breastfeeding women is limited. Given that infants' gut microbiota is still developing and polyphenols may affect nutrient absorption and metabolism, moderation is key. Pregnant and breastfeeding women, along with parents introducing polyphenol-rich foods to infants, should focus on a balanced diet. More studies are needed on the long-term effects, optimal dosages, and interactions during critical developmental stages.”

8) Please provide a table listing the human studies on the topic. This will strengthen the proposed argument that polyphenols are indeed useful. Otherwise, it remains speculative whether this would actually work in human.

RESPONSE: Since CKM syndrome is a relatively new term, there is currently limited human research available on this subject. This point has been addressed in the Introduction.

Lines 54-60: “A notable example is the Dutch famine study, which found that maternal undernutrition during pregnancy is associated with a higher risk of adult offspring developing coronary heart disease, kidney disease, obesity, hyperlipidemia, and hypertension—all features of CKM syndrome [11]. However, there is currently limited human research that provides evidence on how maternal insults can lead to all aspects of CKM syndrome in both children and adults, as this is a relatively new and broad field.”

9) CKM is an extremely broad terminology. I think whenever possible, the authors need to specify the details of cardiovascular and kidney diseases explored in each studies and how they induced the disease. For instance, hypertension induced CKD would have different pathophysiology than diabetes induced CKD. Thus, the cellular target of polyphenols may vary.

RESPONSE: We acknowledge the Reviewer’s point about the complex interplay among the features of CKM syndrome and their underlying pathogenesis. However, our review primarily focuses on the reprogramming effects rather than the direct impacts of polyphenols. Therefore, we concentrate on summarizing the current evidence related to how early-life polyphenol intervention might help prevent CKM syndrome programming. We discuss common mechanisms involved, such as oxidative stress, abnormal activation of the renin-angiotensin system (RAS), gut microbiota dysbiosis, and epigenetic modulation. We hope the Reviewer understands and supports this focus of our review.

10) The review would benefit from a schematic drawing explaining how maternal health can induce CKM disease in the offsprings and which part of the cascade polyphenols would exert their potentials.

RESPONSE: We have detailed the polyphenols beneficial for offspring with CKM syndrome in Table 2 and 3 and outlined their protective mechanisms against CKM syndrome in Figure 3. We hope these efforts meet your requirements and eliminate the need for an additional figure.

Round 2

Reviewer 1 Report

Comments and Suggestions for Authors

2nd review of Tain and Hsu “Maternal Polyphenols….”

The authors did a very through job of responding to the comments.  The new version offers a more balanced and nuanced picture of polyphenols in health.  I have just a few minor revisions to suggest to clarify a few points.

Original comment & Response:

In Figure 2, it is incorrect to include the tannin type “proanthocyanidin” under non-flavonoids.  This group of compounds are polymers of the flavan-3-ols such as catechin, epigallocatechin, etc.  The text in lines 169-171 has correct information but the figure is misleading.

RESPONSE: We have redrawn Figure 2 to prevent any potential misinterpretation.

2nd review:  Do not omit the polymeric proanthocyanidins (condensed tannins) from the figure.  Move them to the correct position, as a subtype or example of flavanols.

Original comment & response:

Use less dogmatic language …

RESPONSE:  ….Lines 181-185: “Notably, despite initially promising preclinical data suggesting significant health benefits, certain polyphenols, such as curcumin and resveratrol, have failed to demonstrate the same level of effectiveness or impact in human populations. This discrepancy underscores the need for more rigorous and comprehensive studies to better understand the potential and limitations of these polyphenols.”

2nd review:  The new wording makes it sound like only some polyphenols eg curcumin and resveratrol do not show action.  Could you rephrase as follows: 

Notably, despite initially promising preclinical data suggesting significant health benefits, certain polyphenols, such as curcumin and resveratrol,most studies have failed to demonstrate the same level of effectiveness or impact in human populations. This discrepancy underscores the need for more rigorous and comprehensive studies to better understand the potential and limitations of these polyphenols

2nd review suggestion:  In line 217, change “lifespan” to “half life”, the term normally used for drug lifetime in the body.

Original comment & Response:

What is the source for the claim: “Approximately 90-95% of polyphenols are metabolized by gut microbiota, which are 209 essential for breaking down dietary polyphenols and their phase I/II metabolites” (line 209-210).  I am not familiar with studies supporting that idea, in fact studies of polyphenols and the gut microbiome are still quite novel and there is much to learn.

RESPONSE: We have rephrased the following sentence and provided the relevant reference.

Lines 225-226: “Approximately 90% of ingested polyphenols reach the large intestine, where they are transformed into bioavailable products by the resident microbiota [60].”

2nd review: Thanks for including the citation.  Could you reword to say “…where they have the potential to be transformed into bioavailable…”

Author Response

Reviewer #1

2nd review of Tain and Hsu “Maternal Polyphenols….”

The authors did a very through job of responding to the comments.  The new version offers a more balanced and nuanced picture of polyphenols in health.  I have just a few minor revisions to suggest to clarify a few points.

RESPONSE: We apologize that our previous response did not adequately address the reviewer’s concerns.

Original comment & Response:

In Figure 2, it is incorrect to include the tannin type “proanthocyanidin” under non-flavonoids.  This group of compounds are polymers of the flavan-3-ols such as catechin, epigallocatechin, etc.  The text in lines 169-171 has correct information but the figure is misleading.

RESPONSE: We have redrawn Figure 2 to prevent any potential misinterpretation.

2nd review:  Do not omit the polymeric proanthocyanidins (condensed tannins) from the figure.  Move them to the correct position, as a subtype or example of flavanols.

RESPONSE: Corrections have been made.

Original comment & response:

Use less dogmatic language …

RESPONSE:  ….Lines 181-185: “Notably, despite initially promising preclinical data suggesting significant health benefits, certain polyphenols, such as curcumin and resveratrol, have failed to demonstrate the same level of effectiveness or impact in human populations. This discrepancy underscores the need for more rigorous and comprehensive studies to better understand the potential and limitations of these polyphenols.”

2nd review:  The new wording makes it sound like only some polyphenols eg curcumin and resveratrol do not show action.  Could you rephrase as follows:

Notably, despite initially promising preclinical data suggesting significant health benefits, certain polyphenols, such as curcumin and resveratrol,most studies have failed to demonstrate the same level of effectiveness or impact in human populations. This discrepancy underscores the need for more rigorous and comprehensive studies to better understand the potential and limitations of these polyphenols

RESPONSE: As suggested, we have rephrased the sentence accordingly.

2nd review suggestion:  In line 217, change “lifespan” to “half life”, the term normally used for drug lifetime in the body.

RESPONSE: Corrections have been made.

Original comment & Response:

What is the source for the claim: “Approximately 90-95% of polyphenols are metabolized by gut microbiota, which are 209 essential for breaking down dietary polyphenols and their phase I/II metabolites” (line 209-210).  I am not familiar with studies supporting that idea, in fact studies of polyphenols and the gut microbiome are still quite novel and there is much to learn.

RESPONSE: We have rephrased the following sentence and provided the relevant reference.

Lines 225-226: “Approximately 90% of ingested polyphenols reach the large intestine, where they are transformed into bioavailable products by the resident microbiota [60].”

2nd review: Thanks for including the citation.  Could you reword to say “…where they have the potential to be transformed into bioavailable…”

RESPONSE: As suggested, we have rephrased the sentence accordingly.

Reviewer 3 Report

Comments and Suggestions for Authors

Thank you for the response.

Author Response

Reviewer #3

Thank you for the response.

RESPONSE: We thank Reviewer #3 for his/her generous support.
